# Observation of metallic electronic structure in a single-atomic-layer oxide

Byungmin Sohn [1,2,4], Jeong Rae Kim[1,2,4], Choong H. Kim [1,2], Sangmin Lee[3], Sungsoo Hahn[1,2], Younsik Kim[1,2], Soonsang Huh[1,2], Donghan Kim[1,2], Youngdo Kim[1,2], Wonshik Kyung[1,2], Minsoo Kim[1,2], Miyoung Kim[3], Tae Won Noh [1,2✉] & Changyoung Kim [1,2✉]

Correlated electrons in transition metal oxides exhibit a variety of emergent phases. When transition metal oxides are confined to a single-atomic-layer thickness, experiments so far have shown that they usually lose diverse properties and become insulators. In an attempt to extend the range of electronic phases of the single-atomic-layer oxide, we search for a metallic phase in a monolayer-thick epitaxial $SrRuO_3$ film. Combining atomic-scale epitaxy and angle-resolved photoemission measurements, we show that the monolayer $SrRuO_3$ is a strongly correlated metal. Systematic investigation reveals that the interplay between dimensionality and electronic correlation makes the monolayer $SrRuO_3$ an incoherent metal with orbital-selective correlation. Furthermore, the unique electronic phase of the monolayer $SrRuO_3$ is found to be highly tunable, as charge modulation demonstrates an incoherent-to-coherent crossover of the two-dimensional metal. Our work emphasizes the potentially rich phases of single-atomic-layer oxides and provides a guide to the manipulation of their two-dimensional correlated electron systems.

[1] Center for Correlated Electron Systems, Institute for Basic Science, Seoul 08826, Korea. [2] Department of Physics and Astronomy, Seoul National University, Seoul 08826, Korea. [3] Department of Materials Science and Engineering and Research Institute of Advanced Materials, Seoul National University, Seoul 08826, Korea. [4]These authors contributed equally: Byungmin Sohn, Jeong Rae Kim. ✉email: twnoh@snu.ac.kr; changyoung@snu.ac.kr

With the size of electronic devices getting ever smaller, continued efforts to use different schemes, such as single-molecule transistors, are being made to overcome the quantum limit. The discovery of graphene in 2004 and later van der Waals material groups suggests a direction of atomically thin two-dimensional (2D) electronics[1–3]. From the fundamental scientific point of view, an ideal 2D system can provide physics distinct from that of three-dimensional (3D) systems. Accordingly, the 2D van der Waals material groups, associated devices[4], heterostructures[5], and Moiré superlattices[6,7] have been the most intensively studied subjects in condensed matter physics in the past 15 years. However, it is still quite difficult to obtain high-quality and large-size flakes of van der Waals materials for practical applications.

On the other hand, the strongly correlated electron system of transition metal oxides possesses numerous exotic electronic phases with physical properties, e.g., high-temperature superconductivity, Mott transition, and ferromagnetism[8–10]. Most of these phases appear in three-dimensionally connected oxide crystals such as perovskite oxides. These exotic properties in principle can be utilized for device applications. In light of the direction in the 2D van der Waals materials research, transition metal oxides with single-atomic-layer thickness can offer functionalities in the nanometer scale by bridging correlated electron systems and the field of 2D materials. However, only a few oxides are known to have van der Waals bonding and can be exfoliated into atomically thin flakes[11,12].

Epitaxial thin film growth can provide an alternative approach to constructing artificial 2D oxides and their devices on a range of lattice-matched oxide substrates[13] or Si wafer[14]. Despite extensive efforts to realize the 2D electronic oxides, most of the ultrathin oxide films exhibit a metal-to-insulator transition in approaching the single-atomic-layer limit[15,16], which is usually attributed to correlation-driven Mott insulating phases or localization effects. Such monotonous behavior detains the functional spectrum of 2D oxides and limits the integration of the intrinsic physical properties into real devices. Therefore, the demonstration of a metallic single-atomic-layer oxide is highly desired to extend the boundary of functionalities and application possibilities for oxide films.

Here, we report the metallic electronic structure of a single-atomic-layer oxide, a monolayer $SrRuO_3$ (SRO) film with a single $RuO_2$ atomic plane. The challenge in the investigation of a single-atomic-layer oxide arises from the film's vulnerability to extrinsic disorders. We employed angle-resolved photoemission spectroscopy (ARPES) measurement to obtain the intrinsic electronic structure of a single unit-cell (uc) SRO. The electronic structure of a monolayer SRO is found to be strongly correlated. We unveil the origin of the correlated phase and demonstrate the tunable electronic phase by charge modulation. This work adds electronic states to the library of single-atomic-layer oxides which has been limited to insulators until now.

## Results

### Fabrication of charging-free ultrathin heterostructures.
In the study of ultrathin oxide films, transport measurements are routinely performed to study the physical properties. However, transport properties can be sensitive to extrinsic effects such as disorder[17]. Such extrinsic effects tend to be more pronounced for films with reduced thicknesses due to the increased scattering from interfaces and surfaces[18]. More importantly, transport measurements require a conducting path in the film over a macroscopic scale, which is not generally guaranteed. Particularly, the conducting channel can be broken near the step-edges of substrates[19]. In that regard, ARPES, which detects electrons with periodic motion and does not require macroscopic connectivity,

can be an effective tool to study the intrinsic electronic property of ultrathin oxide films.

There have been several ARPES studies on ultrathin epitaxial transition metal oxide films[20–24]. For the case of $SrIrO_3$ and $LaNiO_3$, the studies have even reached the monolayer limit, for which insulating electronic structures have been observed. Here, we note the similarity between monolayer oxide films and their quasi-two-dimensional counterparts; both $Sr_2IrO_4$[25] and $NdSrNiO_4$[26] are antiferromagnetic insulators. Based on this analogy, we chose a single-atomic-layer-thick SRO film (Fig. 1a) whose electronic ground state is currently under debate[27,28]. The monolayer SRO can be also regarded as a two-dimensional analog of a metallic single layer perovskite $Sr_2RuO_4$ which has been intensively studied for its unconventional superconductivity[29] and recently for magnetism[30].

We grew high-quality and charging-free ultrathin SRO heterostructures with various thicknesses. Figure 1b shows a schematic of the SRO heterostructure which consists of 4 uc SRO layer, 10 uc $SrTiO_3$ (STO) buffer layer, and n uc ultrathin SRO layer (n uc SRO), sequentially grown on an STO (001) substrate. In ARPES measurements on ultrathin films, escaping photoelectrons can cause a charging effect that distorts the measured spectra. By introducing a current path of 4 uc-thick SRO layer (conducting layer), we successfully removed the charging effect in the measurements of ultrathin SRO (Fig. 1c) [See "Supplementary Fig. 4" for comparison of the ARPES data with and without the conducting layer]. The 10 uc STO buffer layer (4 nm thickness) decouples the electronic structure of the topmost ultrathin SRO from that of the conducting layer [See "Supplementary Fig. 7" for details].

Scanning transmission electron microscopy (STEM) provides atomic-scale visualization of our charging-free ultrathin SRO heterostructures. Figure 1d displays a cross-sectional high-angle annular dark-field STEM (HAADF-STEM) image with STO [100] zone axis. To image the monolayer SRO, we deposited an additional 10 uc STO capping layer to protect the monolayer SRO. The conducting layer, buffer layer, monolayer SRO, and capping layer are well organized in the preferred order. In the plot of HAADF-STEM intensity across the monolayer SRO and adjacent interfaces shown in Fig. 1e, an abrupt Ru peak out of the surrounding Ti peaks is evident. The abrupt SRO/STO interfaces are further corroborated by energy-dispersive X-ray spectroscopy (EDS) analysis in Fig. 1f. Taken together, we confirmed that our ultrathin SRO layers possess lateral uniformity and atomically sharp interfaces, which will allow for a systematic ARPES study.

### Electronic structures of atomically thin SRO.
Using the charging-free heterostructures in Fig. 1b, we examine the electronic ground state of the ultrathin SRO film. We performed in-situ ARPES on heterostructures with n = 0, 1, 2, 3, and 4 uc. Figure 1c shows angle-integrated photoemission spectra of the heterostructures near Γ. Definite Fermi edges survive down to the monolayer, showing the persistent density of states (DOS) at the Fermi level. A pronounced spectral weight is observed near the Fermi level for 4 and 3 uc samples, consistent with previously reported thick film ARPES results[31]. As the thickness is reduced, the spectral weight shifts toward the high-binding energy side, resulting in weak but definite spectral weight at the Fermi level for the 1 uc film. While the underlying mechanism for the spectral weight transfer needs further investigations, we can clearly observe a metallic electronic ground state for the monolayer SRO.

In order to better understand the metallic behavior of the monolayer SRO film in view of the electronic structure, we investigate the thickness-dependent evolution of the Fermi

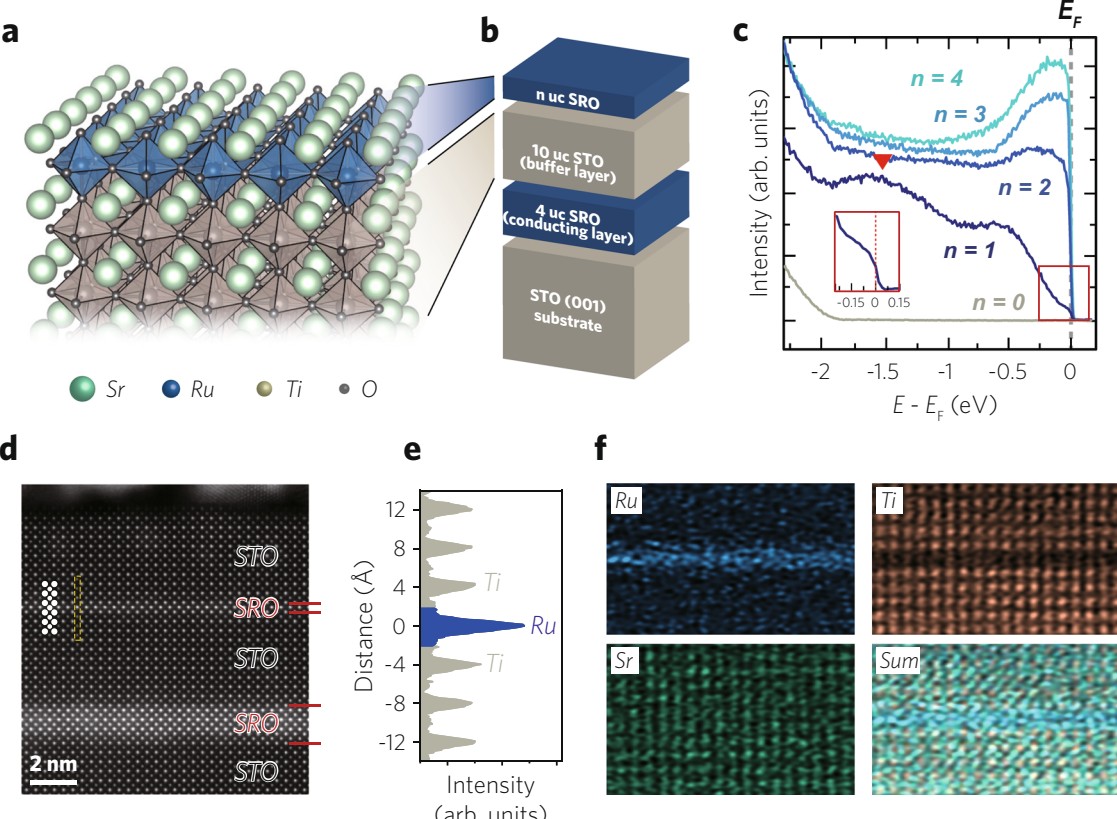

**Fig. 1 Observation of a metallic single-atomic-layer oxide in charging-free ultrathin SrRuO₃ (SRO) heterostructures. a** A schematic of a monolayer SRO grown on a (001)-oriented SrTiO₃ (STO) layer. **b** A schematic of a charging-free ultrathin SRO heterostructure composed of 4 unit-cell (uc) SRO layer (conducting layer), 10 uc STO layer (buffer layer), and *n* uc ultrathin SRO layer, sequentially grown on an STO (001) substrate. The conducting layer prevents the charging effect during the photoemission measurement. The buffer layer decouples the electronic structure of the ultrathin SRO layer from that of the conducting layer. **c** Angle-integrated photoemission spectra from charging-free ultrathin SRO heterostructures. Energy distribution curves (EDCs) are integrated in the range of $-0.6 \ Å^{-1} \leqq k_y \leqq 0.6^{-1}$ and $k_x = 0$. *n* indicates the number of SRO layers. The Fermi edge is persistently seen down to the monolayer. The inset shows a magnified view of the monolayer spectrum near the Fermi level. A hump marked with a red inverted triangle appears at high-binding energy in a monolayer SRO. **d** Atomic-scale imaging of the charging-free monolayer SRO heterostructure obtained by high-angle annular dark-field scanning transmission electron microscopy (HAADF-STEM). **e** HAADF-STEM intensity across the monolayer SRO and adjacent interfaces of the yellow dashed box in (d). **f** Atomic-scale energy-dispersive X-ray spectroscopy (EDS) analysis on the charging-free monolayer SRO heterostructure.

surfaces (FSs). FSs plotted in Fig. 2a are consistent with the angle-integrated photoemission results in Fig. 1c, showing metallic FSs down to the monolayer limit. Here, following the convention used for Sr₂RuO₄[32], we label the three FSs of the $t_{2g}$ bands as $\alpha$, $\beta$, and $\gamma$ as shown in Fig. 2b. The FS maps of 4 and 3 uc films in Fig. 2a show that $\alpha$ and $\beta$ bands are sharp while the $\gamma$ is broad. For the thinner 2 and 1 uc films, $\alpha$ and $\beta$ bands are also slightly broadened. Otherwise, FSs do not show a qualitative change as the thickness varies.

Density functional theory (DFT) calculation of a monolayer SRO in Fig. 2b reproduces the characteristic band structure of experimentally observed FSs [See "Methods" for DFT calculation of a monolayer SRO]. The van Hove singularity (VHS) of the $\gamma$ band is located at the $X$ point. The broad spectral weight at the $X$ point on the $k_x = 0$ line (See Fig. 2a) indicates that the VHS lies close to the Fermi level. We would like to note that the spectral weight near the $X$ point on the $k_y = 0$ line is weak due to the matrix element effect; our ARPES data were measured with He-I$\alpha$ light mostly polarized in the vertical direction[33].

To look at a more detailed thickness-dependent evolution of the $\alpha$ and $\beta$ bands which are composed of $d_{yz}$ and $d_{zx}$ orbitals, we plot in Fig. 2c $\Gamma$-$M$ high-symmetry cut ('Cut' 1 in Fig. 2a) data and momentum distribution curves (MDCs) at the Fermi energy. We observe only two bands for the 2 and 1 uc films despite the

three $t_{2g}$ orbitals. The two dispersive bands are attributed to $\alpha$ and $\beta$ bands located near $k_{\parallel} = 0.6$ and $0.8 Å^{-1}$, respectively, while the $\gamma$ band can be counted out as we will discuss later. The $\alpha$ band, marked with an inverted triangle in the MDC plot in the upper panel, is resolved at the Fermi level regardless of the thickness [See "Supplementary Fig. 5" for the analysis on the $\beta$ band]. Putting it all together, the $\alpha$ and $\beta$ bands are dispersive at all thicknesses but their spectral functions gradually become less coherent as the thickness is reduced.

We now turn our attention to the $\gamma$ band to understand the origin of the thickness-dependent spectral weight transfer to high-binding energy in ultrathin SROs. The VHS of the $\gamma$ band at the $X$ point lies close to the Fermi level which leads to the high DOS (Fig. 3a). Since the high DOS from a VHS can significantly affect the physical properties, it is worth examining the electronic band structures near the VHS. Figure 3b shows the $E$-**k** dispersions along the $k_x = 0$ line, indicated as 'Cut 2' in Fig. 2a. In addition to the $\beta$ band, a coherent heavy $\gamma$ band is clearly observed in the 4 and 3 uc data, whereas the 2 and 1 uc data show only an incoherent $\gamma$ band. As a side note, we attribute the weak $\beta$ band to the matrix element effect as $\alpha$ and $\beta$ bands remain consistently coherent at the Fermi level in the $\Gamma$-$M$ ('Cut 1' in Fig. 2c) and the $\Gamma$-$X$ data along the $k_y = 0$ line [See "Supplementary Fig. 5" for details][33].

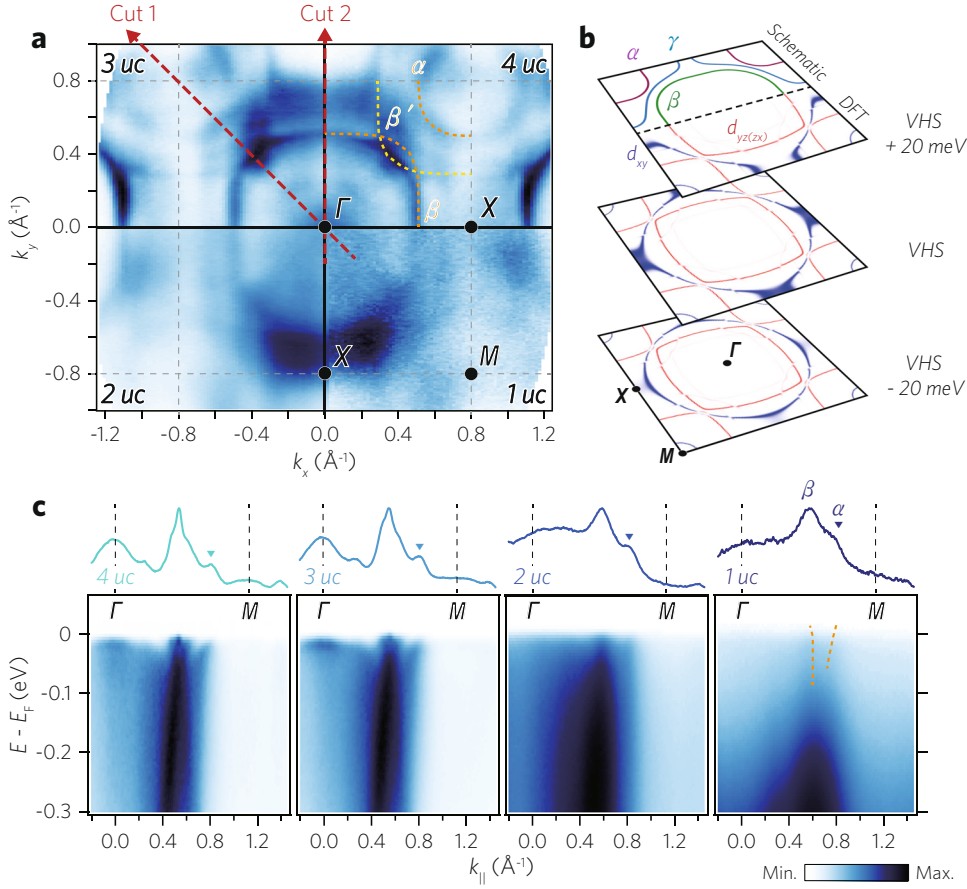

**Fig. 2 Fermi surfaces (FSs) of ultrathin SRO films. a** FSs of ultrathin SRO layers with a specified thickness. Angle-resolved photoemission spectroscopy (ARPES) data were taken at 10 K with He-I$\alpha$ light (21.2 eV) and integrated within $E_F \pm 10$ meV. **b** Energy isosurfaces of two-dimensional (2D) SRO calculated by density functional theory (DFT) calculation, together with schematic energy isosurfaces. DFT results are color-coded for $d_{xy}$ (blue) and $d_{yz,zx}$ (red) orbital contributions to the bands. The VHS represents Van Hove singularity. **c** $\Gamma$-$M$ high-symmetry cuts (Cut 1 in (**a**)) and momentum distribution curves (MDCs) at $E_F$. The inverted triangles next to MDCs represent peaks of the $\alpha$ band. Both $\alpha$ and $\beta$ bands are resolved in 1 uc.

The thickness-dependent evolution of the $\gamma$ band can be better seen in energy distribution curves (EDCs) near the X point, ($k_x$, $k_y$) = (0, 0.75), shown in Fig. 3c. The coherent peak marked by the inverted triangle appears as a kink in the EDCs in the 4 and 3 uc SRO data. On the other hand, the coherent peak disappears in the 2 and 1 uc SRO data, in which the spectral weight is believed to be transferred to the incoherent band. Such behavior is reminiscent of the strongly correlated metallic states in the vicinity of a Mott state as observed in, for example, under-doped cuprates above its superconducting dome[34]. Considering what we learned from the EDCs, it can be deduced that the thickness-dependent electronic transition is closely related to the strong correlation in the $\gamma$ band. Therefore, we may refer to the electronic state of monolayer SRO as an orbital-selectively correlated and incoherent metal. The orbital selective correlated metallic behavior should be closely related to the orbital-selective Mott phase found in $Ca_{2-x}Sr_xRuO_4$[35].

Finally, we discuss the magnetism in these films. While bulk SRO is ferromagnet[36], its quasi-2D analog, $Sr_2RuO_4$, does not show ferromagnetism[29]. It implies a possible magnetic transition in the ultrathin SRO films. We investigated the spin polarization of ultrathin SRO films by performing spin-resolved ARPES (SARPES). Figure 3d shows spin-resolved EDCs measured at 10 K near $\Gamma$. In the high-binding energy region, we observe a sizable difference in the intensity between majority and minority spins for 4 and 3 uc SRO films, indicating ferromagnetism[24]. However, the ferromagnetism is not observed for the 2 and 1 uc SRO films for which no difference is seen in the SARPES data.

**Dimensionality-driven electronic transition**. Note that the different behaviors between in- and out-of-plane orbital bands ascertain the quantum confinement (QC) effect of ultrathin SRO films[37]; the reduced dimensionality affects only the out-of-plane orbital $d_{yz}$ and $d_{zx}$ bands (Fig. 4a). We also notice that there is a remarkable coincidence between the band coherence and ferromagnetism in ultrathin SRO films. Both coherent-to-incoherent and ferromagnetic-to-nonmagnetic transitions start to occur at the 2 uc thickness, which suggests a strong interplay between electronic correlation and magnetism.

In 3D cubic SRO, three $t_{2g}$ orbitals ($d_{xy}$, $d_{yz}$, and $d_{zx}$) are degenerate and each one of them has a VHS. The octahedral rotation, as well as the epitaxial strain, slightly breaks the degeneracy, resulting in a lower energy level for the $d_{xy}$ VHS, as schematically shown in Fig. 4c[38,39]. The QC effect selectively reconstructs the electronic structure of $d_{yz}$ and $d_{zx}$ orbitals. In the 2D monolayer limit, the $d_{yz}$ and $d_{zx}$ bands have strong 1D singularities at the band edges and reduced DOS at the Fermi level ('2D' in Fig. 4c). This reduction in the Fermi-level DOS induces the thickness-driven ferromagnetic-to-nonmagnetic transition between 3 and 2 uc SRO films[37].

The high DOS from the $d_{xy}$ VHS (Fig. 4c) can enhance the effective electron correlation, e.g., Coulomb interaction and Hund's coupling[40]. The instability from the high DOS at the Fermi level can be avoided by splitting the $d_{xy}$ band into spin majority and minority bands, as shown for 4 and 3 uc SRO (left panel, Fig. 4d). However, without the ferromagnetic spin splitting,

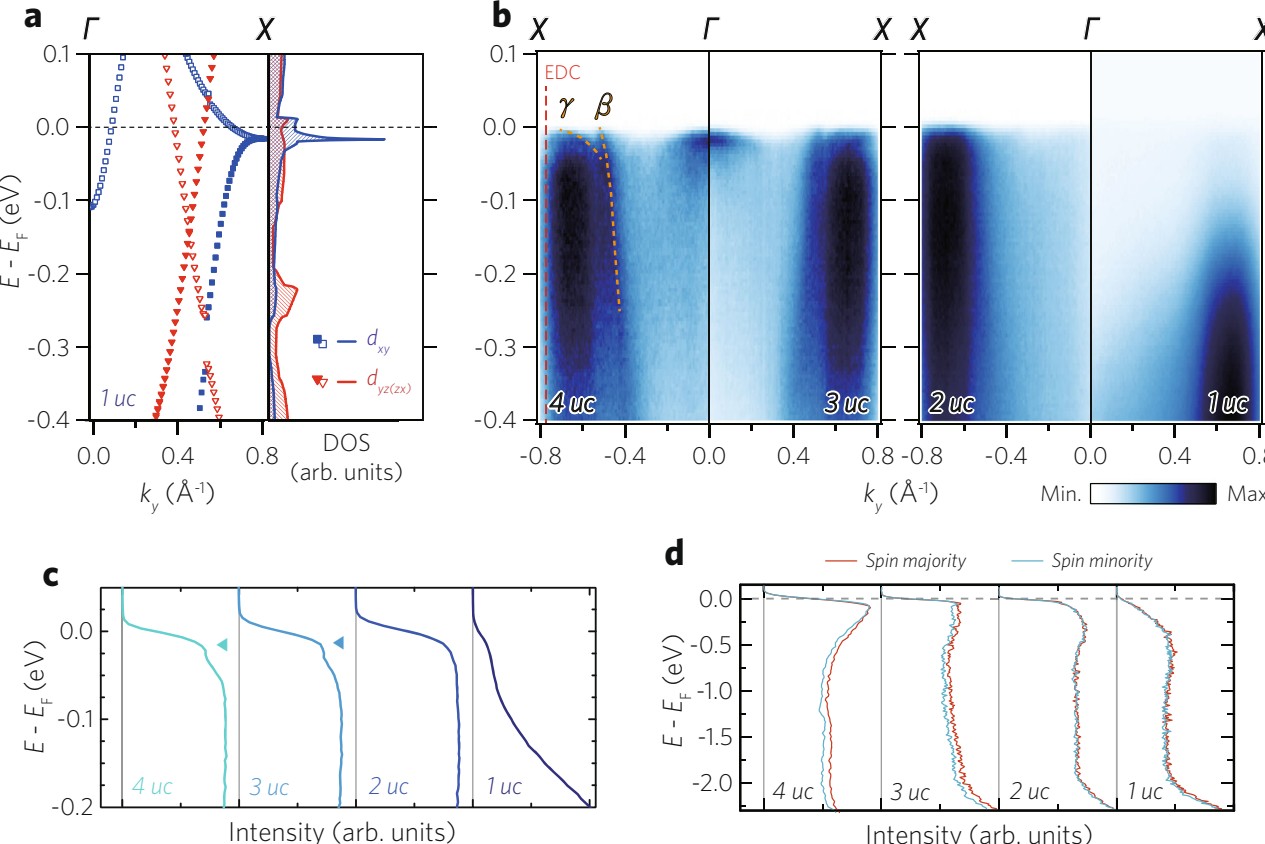

**Fig. 3 Thickness-driven electronic and magnetic transition in ultrathin SRO layers. a** A density functional theory (DFT)-calculated energy versus momentum ($E$-$\mathbf{k}$) dispersion of 2D SRO along $\Gamma$-$X$ high-symmetry line. Open triangles and squares indicate replica bands due to the $\sqrt{2}\times\sqrt{2}$ octahedral rotation. **b** High-symmetry cuts ($\Gamma$-$X$) with a specified thickness of SRO layers along the $k_x = 0$ line. Coherent $\beta$ and $\gamma$ band dispersions are observed in 4 and 3 uc, whereas only an incoherent $\gamma$ is observed in 2 and 1 uc. **c** Thickness-dependent EDCs near the X point, $(k_x, k_y) = (0, \pm 0.75)$, marked with a red dotted line in **b**. Inverted triangles indicate coherent peaks from the $\gamma$ band. The coherent peaks systematically disappear as the thickness is reduced. **d** Thickness-dependent spin-resolved EDCs measured at 10 K near $\Gamma$. The spin majority and minority spectra for 4 and 3 uc films show a difference, showing net spin polarization, whereas the difference vanishes for 2 and 1 uc films.

the $d_{xy}$ band in 1 and 2 uc films retains the VHS, and thus high DOS near the Fermi level. Then, the strong correlation drives the monolayer SRO system to an incoherent-metallic phase (right panel, Fig. 4d). The bilayer SRO case is presumably at the boundary between the coherent ferromagnetic metal and incoherent correlated metal.

**Control of the 2D correlated electronic phase**. With the underlying mechanism for the thickness-driven electronic transition understood, we attempt to exploit the mechanism to control the electronic phase of the monolayer SRO. As the key to the incoherent-metallic phases is the VHS, tuning the chemical potential is likely to break the correlated phase (right panel, Fig. 4d). We used in-situ K dosing to dope electrons into the incoherent-metallic phase of monolayer SRO. Figure 5a shows FS maps of monolayer SRO before and after K dosing. The Fermi wavevector, $\mathbf{k}_F$, of the $\beta$ band changes from 0.52 to 0.60 $\mathring{A}^{-1}$ along the $k_x = 0$ line, indicating that electrons are doped into the system [see "Supplementary Fig. 8" for $E$-$\mathbf{k}$ dispersions and MDCs]. Overall, the K-dosed FS feature is much better resolved in comparison with the result of the pristine case.

The most distinct change occurs in the $\gamma$ band. Figure 5b shows $E$-$\mathbf{k}$ band dispersions along the $k_x = -0.2\,\mathring{A}^{-1}$ line before and after K dosing. The strongly correlated $\gamma$ band has never been sharply resolved in the pristine state but it appears coherent with a clear dispersive feature after K dosing. The $\gamma$ FS became a hole

pocket, which indicates the VHS is now located below the Fermi level (Fig. 2b). This is consistent with our scenario proposed above; the electron doping moves the VHS away from the Fermi level, and consequently, the spectral weight is transferred from the incoherent to coherent peaks. EDCs in Fig. 5c show spectral-weight transfer as well as the emergence of the coherent quasi-particle peak.

We also noticed that the high-binding hump-like peak at $E = -1.5$ eV, which is only observed in monolayer SRO (Fig. 1c), disappears after K dosing as shown in Fig. 5d. A similar hump-like peak at high-binding energy has been reported in photoemission spectroscopy results of CaRuO$_3$ and (Ca, Sr) VO$_3$[41,42] and was attributed to a strong electronic correlation effect. When the Coulomb interaction between electrons increases, a coherent peak near the Fermi level is expected to be suppressed because its spectral weight is transferred to the incoherent lower Hubbard band[42–44]. Thus, we believe that the appearance of the high-binding energy hump with decreasing thickness is also a sign of enhanced electronic correlation in thinner films. The disappearance of the hump-like peak upon K dosing reveals that electron correlations become weaker as the VHS moves away from the Fermi level.

It is worth mentioning a couple of findings from the electron-doping experiments of single-atomic-layer SRO. First of all, we can exclude extrinsic disorders as the origin of the incoherent metallicity in the monolayer SRO. If the incoherent metallicity

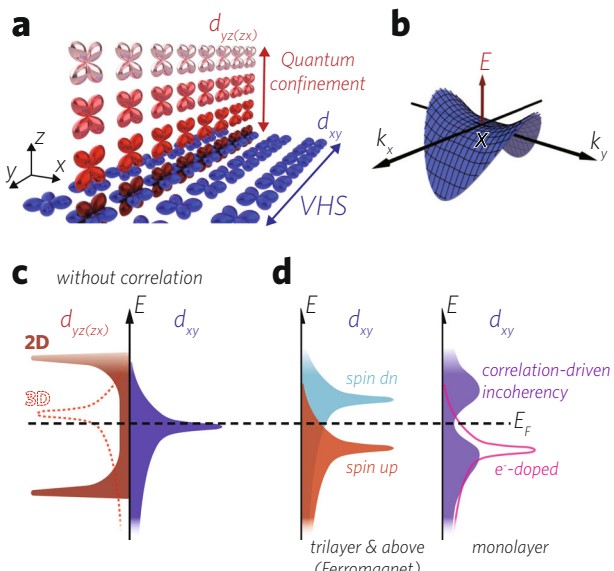

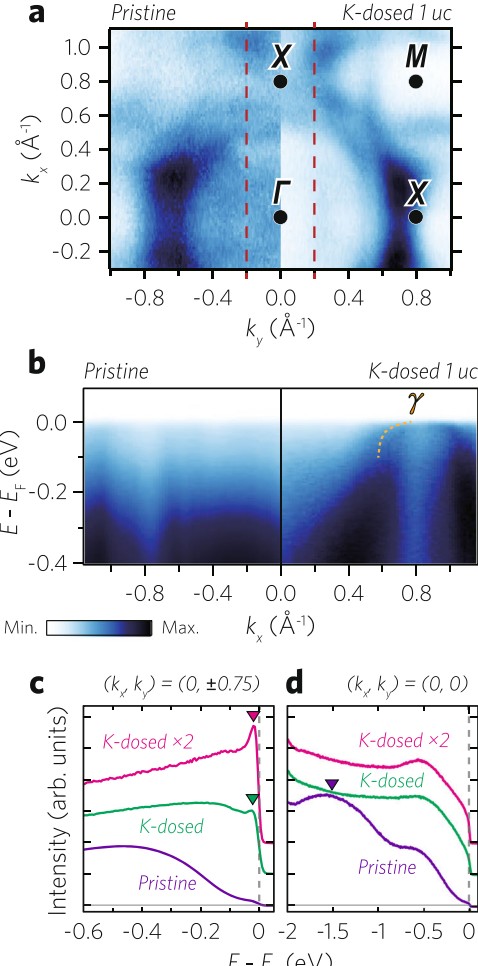

**Fig. 4 Interplay between dimensionality, electronic correlation, and magnetism. a, b** Schematic illustrations of **a** quantum confinement (QC) effect and **b** VHS. The out-of-plane orbitals, $d_{yz}$ and $d_{zx}$, are mainly involved in the QC effect. The in-plane orbital, $d_{xy}$, is responsible for the high density of states (DOS) at the Fermi level due to the VHS. **c, d** Schematic illustration of **c** electronic structures without correlation and **d** thickness-dependent correlation effect on $d_{xy}$ band. The red dotted line represents the $d_{yz}$ and $d_{zx}$ orbital partial DOS for bulk SRO. The high DOS at the Fermi level from the $d_{xy}$ VHS gives rise to a strong electronic correlation. In thick films (more than 3 uc), the DOS at the Fermi level can be significantly reduced via ferromagnetic spin splitting. In the monolayer SRO case, the spin splitting is inhibited due to the QC effect. As a result, the $d_{xy}$ DOS at the Fermi level is reduced via opening a soft gap[35]. When VHS is far from the Fermi level, the coherent states can be recovered.

were due to defects or disorders, it would have been persistent even after K dosing. The reappearance of the coherent peak and disappearance of the high-binding energy hump peak with K dosing support the view for the intrinsic nature of the observed incoherent metallicity of monolayer SRO. The other is that K-dosing results highlight the rich spectrum of electronic phases in monolayer SRO and their giant tunability via charge modulation. The considerable spectral weight at the Fermi energy and clear dispersive bands indicate a good metallicity in the electron-doped monolayer SRO. The incoherent-to-coherent crossover might be exploited to realize atomically thin oxide field-effect-transistors, which have not been realized so far.

2D materials and their applications are a rapidly growing field in contemporary condensed matter physics and materials science. This trend is becoming predominant not only for van der Waals materials but also for oxides. In very recent years, free-standing oxide membranes have been prepared out of the epitaxial oxide heterostructures[45], and the size has reached a wafer scale[46]. Moreover, the high-quality membranes of dielectric oxides are shown to maintain their crystallinity even down to the monolayer limit[47]. By demonstrating a metallic single-atomic-layer oxide, our work expands the scope of 2D oxides that has been limited to insulators so far. The strong electronic correlation gives rise to highly tunable correlated electronic phases, which will be a distinct and advantageous feature of 2D oxides for future research on-device applications. We expect that other emergent phases in oxides, such as unconventional superconductivity[48], could appear in a single-atomic-layer and thus that our findings pave the way to the two-dimensional correlated electronics[49,50].

**Fig. 5 Incoherent-to-coherent crossover in monolayer SRO. a** FS maps of pristine and K-dosed monolayer SROs measured at 10 K. **b** Band dispersions of pristine and K-dosed monolayer SROs along the $k_y = -0.2\,°A^{-1}$ line (red dotted line in **a**). **c, d** EDCs from pristine and K-dosed monolayer SRO films near the X and Γ points normalized by $E = -0.6$ and $-2$ eV, respectively. With K dosing, a quasi-particle peak reappears near the Fermi level, while the hump peak in the high-binding energy region disappears, as marked by inverted triangles. 'K-dosed × 2' indicates twice the dosing amount of 'K-dosed'.

## Methods

**Fabrication of heterostructures**. Epitaxial SrRuO₃ and SrTiO₃ thin films were grown on (001)-oriented SrTiO₃ single crystal substrates by the pulsed laser deposition technique. Prior to the growth, the SrTiO₃ substrate was dipped in deionized water and sonicated for 30 minutes. The substrate was subsequently in-situ annealed in the growth chamber, and the annealing temperature, background oxygen partial pressure, and annealing time were 1,050 °C, $5.0 \times 10^{-6}$ Torr, and 30 min, respectively. Polycrystalline SrRuO₃ and SrTiO₃ targets were ablated using a KrF excimer laser. For the growth of SrRuO₃ films, the substrate temperature, background oxygen partial pressure, and laser energy density were kept at 670 °C, 100 mTorr, and 1.9 J/cm², respectively. For the growth of SrTiO₃ films, the substrate temperature, background oxygen partial pressure, and laser energy density were kept at 670 °C, 10 mTorr, and 1.2 J/cm², respectively. After the growth, all samples have been cooled down to the room temperature with a rate of 50 °C/min in an oxygen partial pressure of 100 mTorr.

**In-situ angle-resolved photoemission spectroscopy**. In-situ ARPES measurements were performed at 10 K using the home lab system equipped with a Scienta DA30 analyzer and a discharge lamp from the Fermi instrument. He-Iα ($hv =$ 21.2 eV) light partially polarized with linear vertical was used. Low-energy electron diffraction patterns are taken after ARPES measurements. Spin polarization was measured with a spin-resolved ARPES system in our laboratory. The system was equipped with a SPECS PHOIBOS 225 analyzer and a very low energy electron diffraction (VLEED) spin detector. For the spin detector, an oxidized iron film

deposited on W(100) was used as the scattering target. He-I$\alpha$ ($h\nu$ = 21.2 eV) light was used as the light source. To clean the surface of SRO thin films, we post-annealed them at 550 °C for 10 mins [See "Supplementary Fig. 9" for details].

**First-principles calculation**. We performed the first-principles calculation using the DFT method without spin-orbit coupling. The PBEsol form of the exchange-correlation functional was used as implemented in VASP[51,52]. To simulate our experimental situation, we prepared a slab geometry with 20 $\AA$ vacuum in which 4 uc of SrTiO$_3$ is sandwiched by 1 uc of SrRuO$_3$ which preserves the mirror symmetry with respect to the middle SrO layer. We used a 600 eV plane wave cut-off energy and $12 \times 12 \times 1$ $k$-points for all calculations and the projector augmented wave method. During the geometry optimizations, the in-plane lattice constant was fixed at the experimental value of SrTiO$_3$, 3.905 $\AA$, and the tolerance on atomic forces was set to 4 meV $\AA^{-1}$. The electronic density of states was calculated using a fine mesh $12 \times 12 \times 1$ $k$-points. For the Fermi surface calculations, we used the PyProcar package[53] to unfold the band structure. The chemical potential in the calculated dispersion was shifted by 70 meV so that $k_F$ of the $\beta$ band coincides with the experimental value.

**Scanning transmission electron microscopy measurement**. Cross-sectional scanning transmission electron microscopy (STEM) specimen was prepared utilizing focused ion beam milling with FEI Helios 650 FIB and further thinned by focused Ar ion milling with Fischione NanoMill 1040. STEM images and energy-dispersive X-ray spectroscopy (EDS) were acquired using Thermo Fisher Scientific Themis Z equipped with a corrector of spherical aberrations, a high-brightness Schottky-field emission gun operated at a 300 kV electron acceleration voltage, and Super-X EDX system. The semi-convergence angle of the electron probe was 25.1 mrad.

## Data availability
The data that support the findings of this study are available from the corresponding author upon reasonable request.

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

## Acknowledgements

We gratefully acknowledge useful discussions with Woo Seok Choi, Kookrin Char, Kee Hoon Kim, Hongki Min, Bohm Jung Yang, Sang Mo Yang, and Seo Hyoung Chang. This work is supported by the Institute for Basic science in Korea (Grant No. IBS-R009-D1, IBS-R009-G2). We acknowledge the support from the Korean government through National Research Foundation (2017R1A2B3011629). Cs-corrected STEM works were supported by the Research Institute of Advanced Materials (RIAM) in Seoul National University.

## Author contributions

B.S., J.R.K., T.W.N. and C.K. conceived the project. B.S. synthesized and characterized the SrRuO$_3$ films. J.R.K. conceived and fabricated the charging-free ultrathin heterostructures. B.S., J.R.K., Youns.K., D.K. and Young.K. conducted ARPES measurements. B.S., Sung.H. and Soon.H. conducted spin-resolved ARPES measurements. C.H.K. carried out first-principle calculations. S.L. performed TEM analysis under the supervision of M.K., B.S. and J.R.K. analyzed ARPES data supported by W.K. and M.K., B.S., J.R.K., T.W.N. and C.K. wrote the paper with contributions from other authors. All authors participated in the discussions and commented on the manuscript.

## Competing interests

The authors declare no competing interests.
