## [Peer Review File · Nature Communications]

Observation of metallic electronic structure in a single-atomic-layer oxideEditorial Note:

Parts of this peer review file have been redacted as indicated to maintain the confidentiality of unpublished data.

REVIEWER COMMENTS

Reviewer #1 (Remarks to the Author):

In this manuscript, by combining the epitaxial growth and angle-resolved photoemission spectroscopy (ARPES), the authors have visualized how the electronic structure of the typical perovskite ruthenate evolves with dimensionality. They successfully observed the metallic phase in the monolayer SrRuO₃ film and revealed that the interplay between both dimensionality and correlations render the monolayer SrRuO₃ an incoherent metal with orbital-selective correlation.

Overall, the manuscript is well written, the data quality is fine, and the conclusions are convincing. Particularly, I consider that this experimental work is an important piece of the methodology in line with the recent development in in-situ electronic structure probing technique. This work is also a vast improvement on previous ARPES studies on ultrathin epitaxial ruthenates, e.g., PRL 110, 087004 (2013) and PRB 93, 121102(R) (2016). Hence, this work potentially deserves to be published in Nature communications. However, before the publication, the authors need to address the following comments and questions:

1) One of my main concerns is the inconsistency between the authors' explanation to dimensionality-driven electronic transition and their experimental results. They have discovered the alpha and beta bands (mainly from dxz/dyz orbitals) have nearly thickness independent dispersions while the gamma band shows pronounced thickness-dependent evolution. However, they claimed in Fig. 3a the reduced dimensionality affects only the out-of-plane orbital dyz and dzx bands. How to understand such phenomena?

2) One other concern is the k_z problem. As a three-dimensional perovskite, bands from dxz/dyz orbitals of SrRuO₃ should be dispersive along c direction, and their measurements

probe only a single value of the out-of-plane momentum. How would this limitation in experiments affect the conclusion? The authors need to assess the impact.

3) As for the spin-resolved ARPES data, the authors demonstrated the EDCs taken near the Gamma point. According to previous reports, these states mainly originate from surface states, and there is no spectra weight in the zone center based on the authors' DFT calculation (Fig. 2b). How can we understand the difference between majority and minority spins in surface bands for four or three uc films? How about bulk bands from dxy orbitals?

4) The authors presented the in-situ photoemission spectra of potassium surface doped monolayer SrRuO₃, which illustrated the control of the 2D correlated electron phase from the correlation-driven incoherent state to the coherent state. It is interesting and necessary to figure out how this crossover occurs in details. Thus, it is better for the authors to show the evolution of photoemission spectra with different potassium doping level, which might provide more insight into this phase transition.

Reviewer #3 (Remarks to the Author):

In this manuscript, a “strongly correlated” metallic state is observed in a single-layer SrRuO₃ film by ARPES. The analysis on the electronic structure shows that the SRO film changes from coherent ferromagnetic metal to incoherent correlated metal as the thickness of the film decreases to a critical thickness of 2 uc. This work enriches the potential phases of single-layer transition metal oxides. But I have a few questions to be addressed :

Question 1 :

In this manuscript, authors claimed that the 10 uc STO buffer layer was used to decouple the upper and lower SRO, but there is no convincing experimental data. The question is how to prove that the decoupling is successful, and how to prove that the detected signal does not contain that of the SRO with 4 uc? It is necessary to provide the ARPES data for the films without SRO at the topmost layer.

I wonder why authors do not directly replace STO substrate with a conductive Nb-STO, and then grow n uc SRO/STO/Nb-STO film to completely avoid the signal from the SRO with 4 uc.

Question 2 :

The transport measurement on 2 uc sample already shows insulating behavior, but the ARPES data shows that 2 uc sample is still metal. Such discrepancy is explained that the extrinsic effects (such as disconnected conducting path at step terraces) make it difficult to obtain the intrinsic transport properties for the ultrathin films, while the HAADF-STEM data has proved that the films possess lateral uniformity and atomically sharp interfaces. Is it contradictory? Moreover, the metallic state in the 2 uc sample is observed by transport measurement in the reference (Phys. Rev. Lett. 103, 057201 (2009)). Authors should improve the quality and uniformity of the film to observe the metallic behavior for a two-layer or even a single-layer sample.

Question 3:

In the paper, the explanation for the transition from ferromagnetic to nonmagnetic with decreasing the thickness of the film is because the QC has strong effect on the d_{yz} and d_{zx} orbitals, which leads to reconstruction of the electronic structure, and consequently reduction of DOS at the Fermi level. Such reduction of DOS at the Fermi level leads to the thickness-driven ferromagnetic-to-nonmagnetic transition between SRO films with 3 and 2 uc. However, authors claimed that the dispersions of α and β bands from d_{yz} and d_{zx} orbitals have nearly nothing to do with thickness. These statements are not consistent with each other.

Question 4:

The discussion on the increase of the correlation with decreasing the thickness is based on theoretical analysis or speculation. Authors should provide some experimental evidence to support their conclusion.

The authors in this work set out to address a question that is of fundamental interest to the condensed matter community specifically the vibrant field of ruthenates – how does the electronic structure and magnetic ground state of SrRuO₃ evolve with film thickness and what is the 1 monolayer (ML) asymptotic limit?

This is a controversial topic, as mentioned in the manuscript, further illustrated by ref. 29 (Boschker H. *et al.*) and the references cited therein. The underlying reason for the discrepancy in published data is the extreme sensitivity of the ground state of ruthenates to disorder. Here is a non-exhaustive list of examples -- 100s of ppm level of impurities can suppress superconductivity in Sr₂RuO₄ due to elastic scattering (A.P. Mackenzie *et al.*, Phys. Rev. Lett. **80**, 161 (1998)), Ru-deficiency can reduce the ferromagnetic T_c of SrRuO₃ (B. Dabrowski *et al.*, Phys. Rev. B. **70**, 014423 (2004)) and the sensitivity to disorder of the magnetic field tuned nematic phase in Sr₃Ru₂O₇ (S.A. Grigera *et al.*, Science **306**, 1154 (2004)).

Although, the manuscript is well written and the ARPES measurements carefully conducted and the spectra carefully analyzed, the quality of the few ML SrRuO₃ layer is questionable. I will outline my concerns here in the following bullet points.

- A well-established method of characterizing the quality of SrRuO₃ thin films is using residual resistivity ratio (RRR) which is defined as the ratio of the resistivity (or resistance) of the thin film at room temperature to the resistivity at 4 K or lower. The idea is that the resistivity at low temperatures will be dominated by elastic scattering from disorder, so the lower the disorder the higher the RRR. The authors show the resistivity vs temperature curves for 2, 3 and 4 ML SrRuO₃ layers in Fig. S4 and clearly explain that the RRR for such thin layers will be dominated by interface and surface scattering. The authors should, however, also show the R vs T curve for a thick (say 20 nm or thicker) SrRuO₃ layer grown under the same growth conditions as the 1, 2, 3 and 4 ML layers used for the ARPES studies. This will provide a baseline for the film quality using a metric that is well recognized in the community. I am sure such films are available which were probably grown during the commissioning of the pulsed laser deposition (PLD) system used for this study.
- The broadness of the quasiparticle (QP) peak in the energy distribution curves EDCs in Fig. 1c, even for the 4 ML layer, again calls into question the quality of the films. This is illustrated by Fig. 1d of ref. 32 (D. Shai *et al.*, Phys. Rev. Lett. **110**, 087004 (2013)) where the authors of that manuscript show that the broadness of the QP peak in the EDCs is inversely correlated with the RRR of the films being measured. If possible, the authors of the current manuscript should show the EDC of a thick (say 20 nm or thicker) SrRuO₃ layer to show that the quality of the thicker layers are comparable to what has been published in literature.
- The main argument, made by the authors, that the 1 ML SrRuO₃ layer is disorder free is based on their K-dosing studies. I will admit that I am not an expert on K-dosing, but based on what I have read in literature and conversations with ARPES experts, K-dosing usually makes all the spectral features broader due to the additional surface scattering of the photoemitted electrons from the K-layer. Looking closely at Fig. 5b there seems to be a reduction in the intensity of the spectra on the right panel, ie, the K-dosed 1 ML spectra seems to be more 'noisy' which would support the claim the scattering from the surface K-layer is reducing

the photoelectron count. But at the same time Fig. 5a shows the emergence of a Fermi surface which would support the argument that electron doping from K-dosing does indeed shift the Fermi level away from the vHS of the γ -band. Then again even after K-dosing the QP peak in Fig. 5c is not very sharp. Perhaps the authors could show the K-dosed and pristine Fermi surface of the 1 ML SrRuO₃ sample side-by-side at the same saturation. Any further data or clarifications that the authors can provide to further support this section on K-dosing will go a long way in convincing readers and reviewers that the observations in this paper are indeed intrinsic and not disorder-induced.

- Without establishing clearly that the 1 ML SrRuO₃ layer is disorder free it is hard to accept the main conclusions of this paper, namely, the absence of spin splitting of the EDCs is due to dimensionality induced crossover to a non-ferromagnetic ground state and a dimensionality driven emergence of an incoherent metallic ground state.
- The authors should provide more information in the methods section on how the samples were cooled down at the end of growth. The biggest challenge with growing stoichiometric SrRuO₃ (or for that matter any ruthenates) is the volatility of higher oxides of ruthenium such as RuO₃ and RuO₄ as outlined in W. Siemons *et al.*, Phys. Rev. B **76**, 075126 (2007) and numerous other publications on this topic. In order to minimize the loss of ruthenium was there a supplemental flux of ruthenium supplied to the surface during cool down from growth temperature? This is especially critical for the 1 ML sample. Were the samples annealed in vacuum or in an oxygen ambient? What is the reason for doing this post growth anneal as one would expect the surface to be clean and free of contaminants at the end of PLD growth.

We would like to thank all the reviewers for their careful reading of our manuscript and constructive suggestions. We did our best to address all the issues raised by the reviewers and improved our manuscript based on their comments/suggestions. Below we give a briefly summary of the main changes of our manuscript, followed by the detailed responses to each reviewer's specific comments.

List of main changes made

Main changes in the revised manuscript are listed below.

(1) Author order changes: 'B. Sohn, J. R. Kim, C. H. Kim, S. Lee, S. Hahn, S. Huh, D. Kim, Y. Kim, W. Kyung, M. Kim, **Y. Kim**, M. Kim, T. W. Noh, and C. Kim' → 'B. Sohn, J. R. Kim, C. H. Kim, S. Lee, S. Hahn, **Y. Kim**, S. Huh, D. Kim, Y. Kim, W. Kyung, M. Kim, M. Kim, T. W. Noh, and C. Kim', since Y. Kim helped to perform K-dosing ARPES.

(2) A hump-like peak observed in the high-binding energy of a monolayer SRO is marked with an inverted triangle in Fig. 1c. A corresponding sentence is added in the caption of Fig. 1.

(3) A description of DFT results in Fig. 2b has been revised in the caption.

(4) A description for thickness-dependent behavior of d_{xz} and d_{yz} orbitals has been revised in the main manuscript (page 5): from 'Putting it all together, the α and β bands have thickness independent dispersions.' to 'Putting it all together, the α and β bands are dispersive at all thicknesses.'

(5) Figs. 5a, c, and d have been revised. Especially, we newly added a 'K-dosed $\times 2$ ' data for which the amount of potassium on the SRO monolayer is twice the original amount ('K-dosed' in the submitted manuscript). Corresponding descriptions have been revised in the caption of Fig. 5 and the main manuscript.

(6) A paragraph to explain the appearance and disappearance of a hump-like peak observed in the high-binding energy of a monolayer SRO has been added in the main manuscript (page 6-7).

(7) Several figures have been added and revised in the Supplementary Materials.

- 'Figure S3. Characterization of SRO thin films' has been revised.
- 'Figure S6. Photoemission spectra from SRO heterostructures' has been added.
- 'Figure S7. Band dispersions of STO(n uc)/SRO(4 uc)/STO substrate' has been added.
- 'Figure S8. Post-annealing-temperature-dependent photoemission spectra from 20 uc SRO thin films on a STO (001) substrate near the Γ point' has been added.

(8) We have gone through sincere efforts to revise the manuscript in order to prevent grammatical errors, making the manuscript more readable and explicit.

For specific comments, please refer to the detailed replies to the comments below.

Reply to comments from Reviewer #1:

- (1) *“In this manuscript, by combining the epitaxial growth and angle-resolved photoemission spectroscopy (ARPES), the authors have visualized how the electronic structure of the typical perovskite ruthenate evolves with dimensionality. They successfully observed the metallic phase in the monolayer SrRuO₃ film and revealed that the interplay between both dimensionality and correlations render the monolayer SrRuO₃ an incoherent metal with orbital-selective correlation.*

Overall, the manuscript is well written, the data quality is fine, and the conclusions are convincing. Particularly, I consider that this experimental work is an important piece of the methodology in line with the recent development in in-situ electronic structure probing technique. This work is also a vast improvement on previous ARPES studies on ultrathin epitaxial ruthenates, e.g., PRL 110, 087004 (2013) and PRB 93, 121102(R) (2016). Hence, this work potentially deserves to be published in Nature communications. However, before the publication, the authors need to address the following comments and questions:”

Authors’ response: We thank the reviewer for acknowledging the importance of our work and recommending our manuscript for publication. As the reviewer mentioned, we successfully observed the metallic phase in monolayer SrRuO₃ films and revealed the relationship between dimensionality and correlations. Moreover, we also believe that the *in-situ* electronic structure measurement method we developed will help to resolve some of the issues associated with phase transitions in other oxide ultrathin films. Here, we reply to the reviewer’s detailed comments.

- (2) *“1) One of my main concerns is the inconsistency between the authors’ explanation to dimensionality-driven electronic transition and their experimental results. They have discovered the alpha and beta bands (mainly from dxz/dyz orbitals) have nearly thickness independent dispersions while the gamma band shows pronounced thickness-dependent evolution. However, they claimed in Fig. 3a the reduced dimensionality affects only the out-of-plane orbital dyz and dzx bands. How to understand such phenomena?”*

Authors’ response: We would like to thank the reviewer for pointing this out. In our experimental results, the α and β bands appear to have little thickness dependence, while γ band shows pronounced thickness-dependent evolution as shown in Figs. 2 and 3(b) in the main manuscript. This appears to be inconsistent with our claim of the dimensionality-driven transition. We have to admit that the description in the submitted version of the manuscript was somewhat confusing on this issue. In fact, we were trying to describe the changes in the electronic structure from two perspectives: thicknesses dependent (1) ‘band structure change’ and (2) ‘electronic correlation’.

First, we would like to describe **(1) thickness dependent ‘band structure change’**. The d_{yz} and d_{zx} orbitals have prominent interlayer interaction since they are out-of-plane orbitals, while the d_{xy} orbital has a negligible interlayer interaction due to its in-plane character. As a result, thickness variation should mostly affect the dispersion of the d_{yz} and d_{zx} bands, in the form of band splitting while the dispersion of the d_{xy} band remains more or less the same. This is what we intended to explain with the Figs. 4(a) and (c), and also is explained in Y. Chang et al. [PRL 103, 057201 (2009)] and B. Sohn et al. [arXiv:1912.04757].

One should expect to observe the corresponding change in the α and β (d_{yz} and d_{zx}) bands in ARPES data. For example, Fig. A1 shows DFT calculated Fermi surface of mono- and four-layer SROs. The two systems have different band dispersions as shown in Fig. A1(a). However, the change is not clearly observed in actual ARPES data for an intricate reason which is discussed in detail in B. Sohn et al. [arXiv:1912.04757]. First of all, the large splitting between d_{yz} and d_{zx} bands due to the interlayer coupling pushes many of them away from the Fermi energy. In addition, the electronic-correlation significantly broadens the bands in the high-binding energy region, making them almost unobservable. As a result, the measured α and β Fermi surfaces of mono- and four-layer SROs look quite similar even though the underlying band structures are different.

The remaining question is then why the γ or d_{xy} band pocket has strong thickness dependence (Fig. 3(b) in the main manuscript). It is where the (2) *thickness dependent ‘electronic correlation’* plays the role. As d_{xy} orbital is an in-plane orbital, the thickness cannot directly affect the γ band dispersion as we explained above. We assert that the change in the γ band stems from the VHS of the d_{xy} band. Due to the high density of states (DOS) near the VHS of the γ band, correlation-driven incoherence is more pronounced in the γ band than in the α and β bands, i.e., the change induced by electronic correlation. We believe the K-dosing data support our claim. Indeed, these are consistent with our experimental observation in which the d_{yz} and d_{zx} bands are dispersive, whereas the d_{xy} band are strongly incoherent in the bi- and mono-layer limit.

We believe that the sentence, “... the α and β bands have roughly thickness independent dispersions ...” in our main manuscript, can be misleading. We accordingly revised the manuscript.

Fig. A1. (a) DFT band structures of mono- (upper) and four-layer (lower) SrRuO₃ (SRO). Fermi surfaces of the spin down bands of (b) single-layer and (c) four-layer SRO slabs. From [B. Sohn et al., arXiv:1912.04757]

- (3) “(2) One other concern is the k_z problem. As a three-dimensional perovskite, bands from dxz/dyz orbitals of SrRuO₃ should be dispersive along c direction, and their measurements probe only a single value of the out-of-plane momentum. How would this limitation in experiments affect the conclusion? The authors need to assess the impact.”

Authors’ response: We thank the reviewer for the comment. As the reviewer mentioned, a three-dimensional material is expected to have three-dimensional electronic structure, showing dispersive band structure along the k_z direction. On the other hand, as we study ultrathin films, no k_z dispersion but band splitting due to inter-layer coupling is expected. Therefore, using a single photon energy (therefore accessing only a single k_z) should not affect the conclusion.

k_z independent electronic structure of ultrathin SRO film has been investigated by ARPES measurement on a 4 uc SRO thin film [B. Sohn et al., arXiv:1912.04757]. Figure A2 (a) shows a Fermi surface map as well as the energy-momentum (E-k) spectra of a 4 uc SRO thin film measured with 80 eV light with linear horizontal (LH) polarization. A detailed Fermi surface map is shown in Fig. A2 (b) while Fig. A2 (c) shows the Fermi surface in the (k_x , k_z) plane. It is clearly seen in (c) that the β band does not have any k_z dispersion. That is, 4 uc SRO thin films do not have a considerable k_z dependence.

Based on the 4 uc data, it is accordingly reasonable to assume negligible k_z dependence for 1 - 3 uc films. Therefore, we believe that studying the electronic structure of ultrathin films with a single photon energy should not affect the conclusion.

Fig. A2. Photon energy dependent ARPES data of a 4 unit cell (uc) SRO thin film. (a) Intensity plot of ARPES data obtained with linear horizontal (LH) polarized synchrotron light with a photon energy of 80 eV. (b) Fermi surface map of the data in the k_x - k_y plane. (c) Fermi surface map in the k_x - k_z plane (left) and its 2D curvature (right). An inner potential of 14 eV is used. From [B. Sohn et al., arXiv:1912.04757].

- (4) “3) As for the spin-resolved ARPES data, the authors demonstrated the EDCs taken near the Gamma point. According to previous reports, these states mainly originate from surface states, and there is no spectra weight in the zone center based on the authors’ DFT calculation (Fig. 2b). How can we understand the difference between majority and minority spins in surface bands for four or three uc films? How about bulk bands from d_{xy} orbitals?”

Authors’ response: The reviewer states that the EDCs taken near the Gamma point originate from surface states. We believe that the surface states the reviewer #1 mentioned are the surface states of single-crystalline Sr_2RuO_4 , which have been previously reported [for example, A. Damascelli et al., Phys. Rev. Lett. 85, 5194 (2000); W. Kyung et al., npj Quantum Mater. 6, 5 (2021)]. The surface states are known to be induced due to the inversion symmetry breaking on the Sr_2RuO_4 surface, which produces octahedra rotations at the topmost layer of single-crystalline Sr_2RuO_4 [W. Kyung et al., npj Quantum Mater. 6, 5 (2021)]. Since there is no octahedral rotation in the bulk of single-crystalline Sr_2RuO_4 , some replica surface bands are observed near the Γ point due to the unit-cell doubling ($\sqrt{2} \times \sqrt{2}$). These surface and bulk states are well distinguished in the single-crystalline Sr_2RuO_4 system.

For SrRuO_3 , we would like to mention that octahedral rotation already exists in the bulk. In that case, the ‘replica bands’ of Sr_2RuO_4 surface are part of the bulk bands in SrRuO_3 . It is also known that structures with finite octahedral rotation (as in SrRuO_3) do not show surface states as found for Sr_2RhO_4 [B. J. Kim et al., Phys. Rev. Lett. 97, 106401 (2006)]. Therefore, we believe the EDCs taken near the Γ represents bulk states. As a side note, for few-layer-thick ultrathin films, it is also difficult to distinguish surface states from bulk states (or may not be meaningful).

The reviewer puzzled why the DFT result of a monolayer SrRuO_3 in Fig. 2 (b) appears to show no spectral weight near the Γ point. The reason that there appears to be no states near the Γ point in our DFT calculation is matter of the presentation. What we plot in the DFT part of Fig. 2b are color coded by d_{xy} and d_{yz}/d_{zx} orbital contributions to the bands. In fact, the Γ point spectrum is visible in the non-orbital-selective energy-isosurface (Fig. A3). Indeed, this band appears near the Γ point in our calculation of the band dispersion in Fig. A3. We mentioned that our DFT shows orbital contributions to the bands in the main manuscript.

Fig. A3. DFT calculation of non-orbital-selective energy-isosurface of two-dimensional SRO near the van Hove singularity (VHS).

The reason why we chose the Γ point for spin-resolved ARPES is that the spin polarization at high-binding energies does not really depend on the electron momentum. Figure A4 (a) shows Fermi surface of a 4 uc SRO thin film and Fig. A4 (b) shows spin-resolved ARPES results at high-symmetry points, Γ , X, and M [B. Sohn et al., arXiv:1912.04757]. Spin polarization near the Fermi level is different at each point, while it is positive with a constant value of 10 % regardless of the electron momentum in the high-binding energy region. Thus, for the verification of the magnetic ground state at each thickness, we do not need to choose a specific point. We speculate that the angle-independent spin polarization appears in the high-binding energy due to strong correlation effect as reported in the DMFT calculation [M. Kim and B. I. Min Phys. Rev. B 91, 205116 (2015)]. In short, the band structure is incoherent in the high-binding energy region, and thus we can see a constant value of spin polarization regardless of the momentum.

Fig. A4. Home-lab spin-resolved ARPES data of a 4 uc SRO thin film. (a) Fermi surface map in the (k_x, k_y) plane measured at 10 K. (b) Spin-resolved ARPES data of the valence band at Γ , X and M points. Measurements are taken at 10 K ($T < T_c$) and 125 K ($T > T_c$). Lower figures plot the spin polarization, $P = (\mathbb{I}_\uparrow - \mathbb{I}_\downarrow) / (\mathbb{I}_\uparrow + \mathbb{I}_\downarrow)$, at each high symmetry point. Adopted from [B. Sohn et al., arXiv:1912.04757].

- (5) *“4) The authors presented the in-situ photoemission spectra of potassium surface doped monolayer SrRuO₃, which illustrated the control of the 2D correlated electron phase from the correlation-driven incoherent state to the coherent state. It is interesting and necessary to figure out how this crossover occurs in details. Thus, it is better for the authors to show the evolution of photoemission spectra with different potassium doping level, which might provide more insight into this phase transition.”*

[Redacted]

Reply to comments from Reviewer #2:

- (1) *“The authors in this work set out to address a question that is of fundamental interest to the condensed matter community specifically the vibrant field of ruthenates – how does the electronic structure and magnetic ground state of SrRuO₃ evolve with film thickness and what is the 1 monolayer (ML) asymptotic limit? This is a controversial topic, as mentioned in the manuscript, further illustrated by ref. 29 (Boschker H. et al.) and the references cited therein. The underlying reason for the discrepancy in published data is the extreme sensitivity of the ground state of ruthenates to disorder. Here is a non-exhaustive list of examples -- 100s of ppm level of impurities can suppress superconductivity in Sr₂RuO₄ due to elastic scattering (A.P. Mackenzie et al., Phys. Rev. Lett. 80, 161 (1998)), Ru-deficiency can reduce the ferromagnetic T_c of SrRuO₃ (B. Dabrowski et al., Phys. Rev. B. 70, 014423 (2004)) and the sensitivity to disorder of the magnetic field tuned nematic phase in Sr₃Ru₂O₇ (S.A. Grigera et al., Science 306, 1154 (2004)).*

Although, the manuscript is well written and the ARPES measurements carefully conducted and the spectra carefully analyzed, the quality of the few ML SrRuO₃ layer is questionable. I will outline my concerns here in the following bullet points.”

Authors' response: We thank the reviewer for acknowledging the quality of our work and the importance of the issue. The reviewer raises a potential issue with our interpretation and tries to back the point with references. The criticism is that the ground state of a given sample is highly sensitive to disorder and impurities, e.g. superconducting state in Sr₂RuO₄ and ferromagnetic state in SrRuO₃.

We agree that exceptionally low level of disorder is indispensable for some of the transport properties of emergent quantum phases of ruthenates such as superconductivity in Sr₂RuO₄. We make it clear that, even though our SrRuO₃ films are of high quality with very clean surfaces, the overall disorder & impurity level may not be low enough to realize some of the transport properties.

On the other hand, we respectfully disagree with the reviewer's view that the quality of our films is questionable for the properties we discuss. For example, we would like to point out that the exceptionally low level of disorder is not required to get ARPES data needed for the discussion of metallicity. In fact, in the case of Ca_{2-x}Sr_xRuO₄, the superconductivity disappears immediately as Ca replaces Sr. However, ARPES studies to address the metal insulator transition have been successfully performed across the whole phase diagram (for all values of x) [D. Sutter et al., Nat. Commun. 8, 15176 (2017); A. Shimoyamada et al., Phys. Rev. Lett. 102, 086401 (2009); M. Neupane et al., Phys. Rev. Lett. 103, 097001 (2009); M. Kim et al., arXiv:2102.09760]. This applies to the cases of the ferromagnetic and incoherent metallic ground state of 1 uc SRO. Please refer to our detailed replies below, especially to comment #5.

- (2) *“A well-established method of characterizing the quality of SrRuO₃ thin films is using residual resistivity ratio (RRR) which is defined as the ratio of the resistivity (or resistance) of the thin film at room temperature to the resistivity at 4 K or lower. The idea is that the resistivity at low temperatures will be dominated by elastic scattering from disorder, so the lower the disorder the higher the RRR. The authors show the resistivity vs temperature curves for 2, 3 and 4 ML SrRuO₃ layers in Fig. S4 and clearly explain that the RRR for such thin layers will be dominated by interface and surface scattering. The authors should, however, also show the R vs T curve for a thick (say 20 nm or thicker) SrRuO₃ layer grown under the same growth conditions as the 1, 2, 3 and 4 ML layers used for the ARPES studies. This will provide a baseline for the film quality using a metric that is well recognized in the community. I am sure such films are available which were probably grown during the commissioning of the pulsed laser deposition (PLD) system used for this study.”*

Authors' response: We would like to thank the reviewer for the suggestion. As the reviewer mentioned, information on SRO film characterization can provide a baseline for the film quality. In fact, we have characterized SRO thick films prior to growth of ultrathin films. Figure B1 (a) shows temperature dependent resistivity curves for 2, 3, 4 and 50 uc SRO thin films. The 50 uc (~ 20 nm) SRO thin film shows a residual resistivity ratio (RRR) of 11.33. This value is comparable to or higher than reported values of PLD-grown SRO films but lower than those prepared by molecular beam epitaxy (MBE) [H. P. Nair, *et al.*, APL Materials **6**, 046101 (2018)]. Figure B1 (b) shows the structural characterization of 50 nm thick SrRuO₃ films by X-ray diffraction. The SrRuO₃(002) peak is located at $2\theta = 45.89$, which is slightly smaller than that for optimal MBE films ($2\theta = 45.96$) [W. Siemons, *et al.*, Phys. Rev. B **76**, 075126 (2007)]. This tendency is consistent with the reported relation between RRR and lattice expansion.

Fig. B1. (a) Temperature dependent resistivity for 2, 3, 4 and 50 unit-cell (uc) SrRuO₃ (SRO) thin films. The 50 uc SRO thin film exhibits a high residual resistivity ratio (RRR) value of 11.33. The 4 and 3 uc SRO films show metallic behavior at high temperature, whereas 2 uc sample shows only insulating behavior at any measured temperature. We believe that extrinsic effects such as disconnected conducting path at step terraces make it difficult to measure the intrinsic transport properties of ultrathin films. (b) X-ray diffraction 2θ -theta scan of 50 nm thick SRO film grown on SrTiO₃ (001) substrate.

[Redacted]

We added Fig. B1 in the revised Supplementary Materials.

[redacted]

(3) “The broadness of the quasiparticle (QP) peak in the energy distribution curves EDCs in Fig. 1c, even for the 4 ML layer, again calls into question the quality of the films. This is illustrated by Fig. 1d of ref. 32 (D. Shai et al., Phys. Rev. Lett. 110, 087004 (2013)) where the authors of that manuscript show that the broadness of the QP peak in the EDCs is inversely correlated with the RRR of the films being measured. If possible, the authors of the current manuscript should show the EDC of a thick (say 20 nm or thicker) SrRuO₃ layer to show that the quality of the thicker layers are comparable to what has been published in literature.”

Authors’ response: As the review noted, sharpness of a QP peak can be often regarded as indicator of the sample quality and thus reliability of ARPES data. However, as we will explain, the judgement of the quality of our films based on the comparison between our data and that of D. Shai et al. is misleading.

Fig. B3. Energy-distribution photoemission spectra from SRO heterostructures. (a) Angle-integrated photoemission spectra integrated over the range of $|k_y| < 0.6 \text{ \AA}^{-1}$ and $k_x = 0 \text{ \AA}^{-1}$ (Fig. 1 (c) in the main manuscript). (b) Energy distribution curves (EDCs) of 4, 3, 2, and 1 uc SRO near the Γ point ($|k_y| < 0.05 \text{ \AA}^{-1}$ and $k_x = 0 \text{ \AA}^{-1}$). Clear quasiparticle peaks are observed in 4 and 3 uc SRO. (c) EDCs of 50 and 4 uc SRO near the Γ point ($|k_y| < 0.05 \text{ \AA}^{-1}$ and $k_x = 0 \text{ \AA}^{-1}$). The intensity of quasiparticle peak becomes weaker in 50 uc SRO thin film.

First, we would like to let the reviewer know that the energy distribution curves (EDCs) in Fig. 1 (c) in our main manuscript (Fig. B3 (a)) are integrated over a different range in the momentum space from those in Fig. 1 (d) of ref. 32 [D. E. Shai et al., Phys. Rev. Lett. 110, 087004 (2013)]. The reported EDCs were integrated near the Γ point. For comparison between our data and ref. 32, EDCs near the Γ point are given in Fig. B3 (b). Clear quasiparticle peaks are observed in 4 and 3 uc SRO thin films. As reviewer #2 mentioned, we also performed ARPES on a 50 uc (~ 20 nm) SRO thin film. Figure B3 (c) shows EDCs of 50 and 4 uc SRO thin films integrated near the Γ point. Although the RRR is much higher for the 50 uc SRO film, the quasiparticle peak is smaller for the 50 uc SRO film. We added Fig. B3 in the Supplementary Materials.

[Redacted]

In summary, although our samples have lower RRR values, sharp QP peak could be obtained from our SRO films, corroborating the reliability of are experimental data.

- (4) *“The main argument, made by the authors, that the 1 ML SrRuO₃ layer is disorder free is disorder-free is based on their K-dosing studies. I will admit that I am not an expert on K-dosing, but based on what I have read in literature and conversations with ARPES experts, K-dosing usually makes all the spectral features broader due to the additional surface scattering of the photoemitted electrons from the K-layer. Looking closely at Fig. 5b there seems to be a reduction in the intensity of the spectra on the right panel,*

ie, the K-dosed 1 ML spectra seems to be more 'noisy' which would support the claim the scattering from the surface K-layer is reducing the photoelectron count. But at the same time Fig. 5a shows the emergence of a Fermi surface which would support the argument that electron doping from K-dosing does indeed shift the Fermi level away from the vHS of the γ -band. Then again even after K-dosing the QP peak in Fig. 5c is not very sharp. Perhaps the authors could show the K-dosed and pristine Fermi surface of the 1 ML SrRuO₃ sample side-by-side at the same saturation. Any further data or clarifications that the authors can provide to further support this section on K-dosing will go a long way in convincing readers and reviewers that the observations in this paper are indeed intrinsic and not disorder-induced."

Authors' response: We thank the reviewer for the helpful comments. First of all, we would like to make it clear that we are not claiming our monolayer SRO is disorder-free. What we can say is that the effect of disorder is small enough to make our conclusion. As the reviewer mentioned, K-dosing usually may cause broadening of spectral features due to the additional scattering from the K atoms that photoemitted electrons experience. Therefore, the observed sharpening in the K-dosed 1 uc SRO thin film stems from the intrinsic incoherent-to-coherent metal transition. It is reasonable to assume that the broad spectral weight of the pristine sample is not coming from disorder, because the presumably more-disordered K-dosed sample show sharp spectral weight.

[Redacted]

The reviewer mentioned about the K-dosing data that “Then again even after K-dosing the QP peak in Fig. 5c is not very sharp. Perhaps the authors could show the K-dosed and pristine Fermi surface of the 1 ML SrRuO₃ sample side-by-side at the same saturation.” For a better understanding, we replaced Fig. 5 in the original main manuscript with Fig. B7. As Reviewer #2 suggested, we compare pristine and K-dosed 1 uc SRO thin film data side-by-side at the same saturation (Fig. B7 (a)). We can clearly observe more coherent band features in the K-dosing data. As mentioned by the reviewer, K-dosing usually increases surface scattering and tend to make the spectra broad. High quality spectra of the K-dosed 1 uc SRO indicate that the surface is reasonably clean (disorder free) despite the additional K layer. Furthermore, incoherent spectra observed from the even cleaner surface of the pristine 1 uc SRO indicates that the incoherence is an intrinsic nature of the electronic phase.

We also would like to mention that, after submission of the original manuscript, we noticed the high-binding hump-like peak at $E = -1.5$ eV in the 1 uc SRO disappears after K dosing (Fig. B7 (d)). The “hump” has been reported in ARPES results of CaRuO₃ [H. F. Yang et al., Phys. Rev. B 94, 115151 (2016)] and (Ca,Sr)VO₃ [K. Maiti et al., Europhys. Lett. 55, 246-252 (2001)]. It is believed to be due to strong electronic correlations. Though the mechanism and origin of the “hump” in SRO should be further studied, we can at least say that the disappearance of the “hump” shows that K dosing affects intrinsic property of 1 uc SRO. We discuss the corresponding part in the revised manuscript.

Fig. B7. Incoherent-to-coherent crossover in monolayer SRO. (a) Fermi surface maps of pristine and K-dosed monolayer SROs measured at 10 K. (b) Band distributions of pristine and K-dosed monolayer SROs along the $k_y = 0.2 \text{ \AA}^{-1}$ (red dotted line in (a)). (c, d) EDCs from pristine and K-dosed monolayer SRO films at the X and Γ points, normalized by $E = -0.6$ and -2 eV, respectively. With K dosing, a quasi-particle peak appears near the Fermi level and the hump peak in the high binding energy region disappears, as marked by the inverted triangle. 'K-dosed $\times 2$ ' indicates twice the dosing amount of 'K-dosed'.

- (5) “Without establishing clearly that the 1 ML SrRuO₃ layer is disorder free it is hard to accept the main conclusions of this paper, namely, the absence of spin splitting of the EDCs is due to dimensionality induced crossover to a non-ferromagnetic ground state and a dimensionality driven emergence of an incoherent metallic ground state.”

Authors’ response: We believe the reviewer’s concern on our interpretation is mainly based on this comment. The reviewer is making two points in this comment; (i) non-ferromagnetic ground state of 1 uc film may not be intrinsic but is due to disorder effect, and (ii) the incoherent metallic ground state may also be due to disorder and thus is not an intrinsic effect. We respectfully disagree with the reviewer on these points as detailed below. Let us answer the second one and then the first.

- (i) Incoherent metallic ground state: We would like to emphasize again that exceptionally high RRR (or completely disorder free sample) is not a necessary condition for high quality ARPES data. If the disorder were responsible for the incoherence of the metallic state, K-dosing (which should add even more disorder as the reviewer noted) would not lead to coherent states with very sharp QP peaks (please see our reply to comment #4 and new figures therein). Therefore, there should be something else that makes the electronic structure incoherent in the pristine system. We believe it is the dimensionality driven incoherent metallic ground state.
- (ii) Non-ferromagnetic ground state: If the non-ferromagnetic ground state were indeed induced by disorder, it would be reasonable to assume that such disorder effect should also affect the magnetic state of thicker films. However, SRO films show $T_C \sim 150$ K for RRR values higher than a few, which is well-documented in the literature [C. L. Chen *et al.*, Appl. Phys. Lett. **71**, 1047 (1997)]. Our experience with SRO films is that being able to do ARPES is generally a much more stringent condition than having a ferromagnetic ground state.

The reviewer mentions the work in PRB 70, 014423 (2004) as an example for a disorder effect on ferromagnetism. However, we note that the films are not even metals (RRR less than 1), indicating disorder is extremely severe. Therefore, it is not an appropriate example for the discussion because our thick films show RRR values over 10.

Our 1 uc films show high quality ARPES data, with very sharp QP peak upon K-dosing, which suggests that the disorder in our 1 uc film is much less than those with low RRR films. Therefore, based on these observations, we believe that the disorder in our 1 uc films is not the culprit of the non-ferromagnetic ground state.

There is also supporting spectroscopic evidence for our conclusion. The magnetic ground state transition (from FM to non-FM) occurs concomitantly with the coherent-to-incoherent spectral weight transfer of the γ band, which is an intrinsic effect as the K dosing experiment shows. If the non-FM transition were an extrinsic effect due to disorder, such spectroscopic behavior including the -1.5 eV hump would not occur.

The reviewer raised a reasonable concern on the role of disorder. In addition to the intrinsic origin, non-ferromagnetic ground state and spectral weight suppression near E_F may have an extrinsic origin (i.e., disorder) as the reviewer suggests. Thus, we may start with these two scenarios on an equal footing. However, all other observations (high RRR over 10, well-resolved ARPES data, appearance of extremely sharp QP peaks upon K-dosing, and the hump structure for 1 uc films) point to an intrinsic reason. Therefore, our observations discussed above and STEM results strongly suggest that the disorder at the SRO/STO interface is not as much as one might have expected. With all these evidence, we hope that the reviewer now agrees with our view.

- (6) “The authors should provide more information in the methods section on how the samples were cooled down at the end of growth. The biggest challenge with growing stoichiometric SrRuO₃ (or for that matter any ruthenates) is the volatility of higher oxides of ruthenium such as RuO₃ and RuO₄ as outlined in W. Siemons et al., Phys. Rev. B 76, 075126 (2007) and numerous other publications on this topic. In order to minimize the loss of ruthenium was there a supplemental flux of ruthenium supplied to the surface during cool down from growth temperature? This is especially critical for the 1 ML sample. Were the samples annealed in vacuum or in an oxygen ambient? What is the reason for doing this post growth anneal as one would expect the surface to be clean and free of contaminants at the end of PLD growth.”

Authors’ response: After the growth, all samples have been cooled down to the room-temperature with a rate of 50°C/min in an oxygen partial pressure of 100 mTorr. The idea of using supplemental ruthenium flux suggested by the reviewer is interesting. However, PLD does not have capability of selectively supplying Ru; we use ceramic SrRuO₃ target to supply constituting atoms. At the end of PLD growth, contrary to the reviewer’s expectation, we find that the surface is not clean. This is the reason we do post-annealing at 575°C in oxygen partial pressure of 1.0E-08 Torr. Figure B8 (a) shows why this post-growth annealing procedure is critical for achieving the sharp QP peak. One may be concerned about sample degradation during this post-growth annealing. Indeed, it is the case when we increase the annealing temperature to 700°C. We calculated the ratio between the intensities of the QP and high-binding region and plot it in Fig. B8 (b). We found that the ratio of QP peak has the largest value when the sample is annealed at 600°C, which shows that the optimal annealing condition for cleaning the surface is near 600°C. We added Fig. B8 in the Supplementary Materials.

Fig. B8. Post-annealing-temperature-dependent energy-distribution photoemission spectra from 20 uc SRO thin films on a STO (001) substrate near the Γ point. (a) Spectra before post-annealing (grey) and after post-annealing at 300 °C (black), 400 °C (red), 500 °C (blue), 600 °C (green), and 700 °C (orange). After annealing, a quasiparticle peak (QP) emerges. The intensity of QP increases with post-annealing temperatures but it eventually becomes weak after annealing at 700 °C. (b) The ratio of QP intensity in (a). r is defined as I_{QP}/I_{HB} , where I_{QP} is the intensity between E_F and $E_F - 0.1$ eV, and I_{HB} the integrated intensity between $E_F - 0.45$ eV and $E_F - 0.55$ eV. After post-annealing, r increases until 600 °C. but decreases again when the SRO film is annealed at 700 °C.

After annealing under the optimum annealing condition, ultrathin SRO films remain intact, as 4 uc SRO films show qualitatively similar transport properties regardless of the post growth annealing (Fig. B9).

Fig. B9. Temperature dependent resistivity for 4 uc SRO thin films. The black curve is taken from the as-grown SRO film and the red curve is from an SRO film with post annealing under the optimized condition (575°C, 1.0E-08 Torr, 10minutes).

Reply to comments from Reviewer #3:

- (1) “In this manuscript, a “strongly correlated” metallic state is observed in a single-layer SrRuO₃ film by ARPES. The analysis on the electronic structure shows that the SRO film changes from coherent ferromagnetic metal to incoherent correlated metal as the thickness of the film decreases to a critical thickness of 2 uc. This work enriches the potential phases of single-layer transition metal oxides. But I have a few questions to be addressed : ”

Authors’ response: We would like to thank the reviewer for acknowledging the importance of our work and recommending publication of our manuscript. We sincerely replied to the comments from Reviewer #3 one by one. Please see the followings.

- (2) “Question 1: In this manuscript, authors claimed that the 10 uc STO buffer layer was used to decouple the upper and lower SRO, but there is no convincing experimental data. The question is how to prove that the decoupling is successful, and how to prove that the detected signal does not contain that of the SRO with 4uc? It is necessary to provide the ARPES data for the films without SRO at the topmost layer. I wonder why authors do not directly replace STO substrate with a conductive Nb-STO, and then grow n uc SRO/STO/Nb-STO film to completely avoid the signal from the SRO with 4 uc.”

Authors’ response: We thank the reviewer for raising an interesting issue. We have shown the integrated photoemission data of a charging-free heterostructure without a topmost SRO layer, *i.e.* STO (10 uc) / SRO (4 uc) / STO substrate, in Fig. 1 (c) in the submitted manuscript. We observed no spectral weight in the energy range between -1.8 eV and Fermi energy for films without the topmost SRO layer (see the $n = 0$ case in Fig. C1 (c) below or Fig. 1 (c) of the main manuscript).

Fig. C1. Observation of a metallic single-atomic-layer oxide in charging-free ultrathin SrRuO₃ (SRO) heterostructures. (a) A schematic of a monolayer SRO grown on a (001)-oriented SrTiO₃ (STO) layer. (b) A schematic of a charging-free ultrathin SRO heterostructure composed of 4 unit-cell (uc) SRO layer (conducting layer), 10 uc STO layer (buffer layer), and n uc ultrathin SRO layer, sequentially grown on a STO (001) substrate. (c) Angle-integrated photoemission spectra from charging-free ultrathin SRO heterostructures. Energy distribution curves (EDCs) are integrated in the range of $|k| < 0.6 \text{ \AA}^{-1}$. n indicates the number of SRO layers.

As the reviewer mentioned, the topmost SRO layer can be coupled with the conducting SRO layer if STO layer is quite thin. To demonstrate it, we performed STO buffer layer thickness dependent ARPES. Figure C2 (a) shows how the measured band structure changes with the thickness of STO layer. Figs. C2 (b) and (c) show energy distribution curves (EDCs) at the Γ point for the three cases. Without STO capping layer (STO (0 uc) on SRO (4 uc)), clear Fermi edge is observed at the Fermi level and spectral weight is observed at all energies.

When a 1 uc STO layer is capped on 4 uc SRO, the spectral weight within the range between -2 eV and E_F significantly decreases, even though the Fermi edge is still observed. When 10 uc STO layer is capped, the spectral weight in the -2 eV and E_F energy range completely vanishes. Based on these observations, we believe that the coupling between 4 uc conducting and the top SRO layers becomes negligible with a few uc of STO buffer layer. Studying the critical thickness for decoupling is of interesting issue and can be studied in future studies.

Fig. C2. Band dispersions of STO (n uc) / SRO (4 uc) / STO (001) substrate. (a) Γ -X high-symmetry cuts of 0, 1, and 10 uc STO deposited on a 4 uc SRO. Spectral weight near the Fermi level decreases as the thickness of STO layers becomes thicker. (b) EDCs of 0, 1, and 10 uc STO deposited on a 4 uc SRO near the Γ point (red dotted line in (a)). (c) Enlarged plot of (b). Spectral intensity decreases as STO layers become thicker and completely disappears for 10 uc STO layer.

[Redacted]

In the revised Supplementary Materials, we added the ARPES data measured on STO layers (Fig. C2).

[Redacted]

- (3) *“Question 2: The transport measurement on 2 uc sample already shows insulating behavior, but the ARPES data shows that 2uc sample is still metal. Such discrepancy is explained that the extrinsic effects (such as disconnected conducting path at step terraces) make it difficult to obtain the intrinsic transport properties for the ultrathin films, while the HAADF-STEM data has proved that the films possess lateral uniformity and atomically sharp interfaces. Is it contradictory? Moreover, the metallic state in the 2 uc sample is observed by transport measurement in the reference (Phys. Rev. Lett. 103, 057201 (2009)). Authors should improve the quality and uniformity of the film to observe the metallic behavior for a two-layer or even a single-layer sample.”*

Authors' response: We thanks the reviewer for pointing out our lack of explanation on the length scale. Our HAADF-STEM focuses on a region of several tens of nanometers to assure the atomic-scale analysis on the heterostructure. Figure C5 shows a typical atomic force microscopy (AFM) image of our 4 uc-thick SRO film grown on STO (001) substrate. The SRO film has a step-terrace structure and a smooth surface within a terrace region. Accordingly, our atomic-scale HAADF-STEM cannot easily catch the disconnected conducting path near the step-edge. Note that our device for transport measurement has a length scale of ~1 mm, so the extrinsic effect from the step-terrace structure is unavoidable. We added the AFM image of the 4 uc SRO film (Fig. C5) in the Supplementary Materials.

Fig. C5. Atomic force microscopy image of a 4 uc SrRuO₃ film on SrTiO₃ (001) substrate. Step-terrace structure with 0.4 nm step height is present on the surface. The typical terrace width is around 230 nm.

Achieving a metallic behavior in transport for atomically thin SRO films would certainly be a significant breakthrough, especially in the perspective of device physics based correlated electronic phases of complex oxide heterostructures. However, extrinsic effect including the fundamental non-uniformity in the step-terrace width limits transport measurements over a macroscopic length scale of ~mm. We would like to emphasize that the main purpose of our study is that we use ARPES which investigates electronic structures over a microscopic length scale to overcome the extrinsic effects. That is, our focus is on the intrinsic nature of the electronic structures in the atomically thin limit. The method we developed can be remarkably useful for characterizing the electronic phases of the atomically thin oxide films.

We are not claiming high-quality ultrathin oxide film in terms of transport measurements. As far as transport measurements (suggested by the reviewer) go, our heterostructures are subject to the extrinsic effects caused by the non-uniform terrace width, and even the state-of-art quality films grown by molecular beam epitaxy technique cannot assure the metallic ground state of the atomically thin SRO film [H. Boschker, *et al.*, Phys. Rev. X **9**, 011027 (2019)]. We speculate that a significant progress can be made by (1) preparation of micron-scale step-terrace substrates or (2) fabrication of submicron-scale transport devices within a single terrace. These are beyond the scope of present study.

(4) *“Question 3: In the paper, the explanation for the transition from ferromagnetic to nonmagnetic with decreasing the thickness of the film is because the QC has strong effect on the dyz and dzx orbitals, which leads to reconstruction of the electronic structure, and consequently reduction of DOS at the Fermi level. Such reduction of DOS at the Fermi level leads to the thickness-driven ferromagnetic-to-nonmagnetic transition between SRO films with 3 and 2 uc. However, authors claimed that the dispersions of α and β bands from dyz and dzx orbitals have nearly nothing to do with thickness. These statements are not consistent with each other.”*

Authors’ response: We would like to thank the reviewer for pointing this out. The same comment was made by reviewer 1. Here, we reproduce our reply to reviewer 1’s comment #2.

We would like to thank the reviewer for pointing this out. As the reviewer correctly understood, we experimentally observed that α and β bands have nearly thickness independent dispersions, while γ band shows pronounced thickness-dependent evolution as shown in Figs. 2 and 3(b) in the main manuscript. This appears

to be inconsistent with our claim of the dimensionality-driven transition. We have to admit that the description in the submitted version of the manuscript was somewhat confusing on this issue. In fact, we were trying to describe the changes in the electronic structure from two perspectives: thicknesses dependent (1) ‘band structure change’ and (2) ‘electronic correlation’.

First, we would like to describe (1) *thickness dependent ‘band structure change’*. The d_{yz} and d_{zx} orbitals have prominent interlayer interaction since they are out-of-plane orbitals, while the d_{xy} orbital has a negligible interlayer interaction due to its in-plane character. As a result, thickness variation should mostly affect the dispersion of the d_{yz} and d_{zx} bands, in the form of band splitting while the dispersion of the d_{xy} band remains more or less the same. This is what we intended to explain with the Figs. 4(a) and (c), and also is explained in Y. Chang et al. [PRL 103, 057201 (2009)] and B. Sohn et al. [arXiv:1912.04757].

One should expect to observe the corresponding change in the α and β (d_{yz} and d_{zx}) bands in ARPES data. For example, Fig. C6 shows DFT calculated Fermi surface of mono- and four-layer SROs. The two systems have different band dispersions as shown in Fig. C6(a). However, the change is not clearly observed in actual ARPES data for an intricate reason which is discussed in detail in B. Sohn et al. [arXiv:1912.04757]. First of all, the large splitting between d_{yz} and d_{zx} bands due to the interlayer coupling pushes many of them away from the Fermi energy. In addition, the electronic-correlation significantly broadens the bands in the high-binding energy region, making them almost unobservable. As a result, the measured α and β Fermi surfaces of mono- and four-layer SROs look quite similar even though the underlying band structures are different.

The remaining question is then why the γ or d_{xy} band pocket has strong thickness dependence (Fig. 3(b) in the main manuscript). It is where the (2) *thickness dependent ‘electronic correlation’* plays the role. As d_{xy} orbital is an in-plane orbital, the thickness cannot directly affect the γ band dispersion as we explained above. We assert that the change in the γ band stems from the VHS of the d_{xy} band. Due to the high density of states (DOS) near the VHS of the γ band, correlation-driven incoherence is more pronounced in the γ band than in the α and β bands, i.e., the change induced by electronic correlation. We believe the K-dosing data support our claim. Indeed, these are consistent with our experimental observation in which the d_{yz} and d_{zx} bands are dispersive, whereas the d_{xy} band are strongly incoherent in the bi- and mono-layer limit.

We believe that the sentence, “... the α and β bands have roughly thickness independent dispersions ...” in our main manuscript, can be misleading. We accordingly revised the manuscript.

Fig. C6. (a) DFT band structures of mono- (upper) and four-layer (lower) SROs. Fermi surfaces of the spin down bands of (b) single-layer and (c) four-layer SRO slabs. From [B. Sohn et al., arXiv:1912.04757]

- (5) “Question 4: The discussion on the increase of the correlation with decreasing the thickness is based on theoretical analysis or speculation. Authors should provide some experimental evidence to support their conclusion.”

Authors' response: We thank the reviewer for the important suggestion. In the review process, we thoroughly looked through our data again and realized that our experimental data can support the increased correlation in thinner films in two ways: (1) a high binding "hump" peak at $E = -1.5$ eV observed in 1 uc SRO, and (2) spectral weight transfer when the QP peak is suppressed (Figs. C7 (a) and (b)).

The 'hump', which is remarked with a red inverted triangle in Fig. C7 (a), is clearly visible for 1 uc SRO. As shown in Figs. C8 (a) and (b), the 'hump' feature at $E = -1.5$ eV has been observed in CaRuO_3 [H. F. Yang et al., Phys. Rev. B 94, 115151 (2016)] and $(\text{Ca,Sr})\text{VO}_3$ [K. Maiti et al., Europhys. Lett. 55, 246-252 (2001)]. This hump feature is believed to be induced by the strong electronic correlation in the systems. Thus, we believe that the hump feature observed for 1 uc SRO in Fig. C7 is a direct sign of increased electronic correlation with decreased thickness. We revised the main manuscript to explain the 'hump' feature present in 1 uc SRO.

When the Coulomb interaction between electrons increases, quasiparticle (QP) peak near the Fermi level is expected to be suppressed with its spectral weight transferred to the incoherent lower Hubbard band [H. F. Yang et al., Phys. Rev. B 94, 115151 (2016); S.-K. Mo et al., Phys. Rev. Lett. **90**, 186403 (2003); M. Neupane et al., Phys. Rev. Lett 103, 097001 (2009); G. Kotliar and D. Vollhardt, Phys. Today 57, 53 (2004)]. This is schematically explained in Fig. C8 (b) (ii). We thus believe that decreasing QP spectral weight with decreasing thickness is also a sign of enhanced U/W or electronic correlation in thinner films. We added corresponding discussion in the revised main manuscript.

Meanwhile, we also found that the hump feature disappears and QP peak recovers with K dosing, as shown in the revised Fig. 5 (c) in the main manuscript (Fig. C9 (d) below). This result also supports that the K dosing decreases electronic correlations in 1 uc SRO, which is what we claimed in the main manuscript. We believe that this result also shows how the correlation is related to the 'hump feature' and 'vanishing QP peak'. We newly added discussion on these results in the revised main manuscript.

Fig. C7. (a) Photoemission spectra integrated over the range of $|k_y| < 0.6 \text{ \AA}^{-1}$ and $k_x = 0 \text{ \AA}^{-1}$ (Fig. 1 (c) in the main manuscript). A red inverted triangle indicates a 'hump' feature. (b) Spectra from 4, 3, 2, and 1 uc SRO taken over a smaller momentum range near the Γ point ($|k_y| < 0.05 \text{ \AA}^{-1}$ and $k_x = 0 \text{ \AA}^{-1}$). A clear QP peak is observed for 4 and 3 uc SRO and spectral weight transfer of QP is observed with decreasing thickness.

Fig. C8. (a) Integrated EDCs of CaRuO₃ (CRO) (blue curve) and SRO (i), and schematic of CRO (ii) showing a bigger GdFeO₃-type distortion than SRO (iii). (b) Pronounced spectral weight transfer in a Mott-Hubbard system Sr(Ca)VO₃, reproduced from K. Maiti et al., [Europhys. Lett. 55, 246-252 (2001)] (i). Schematic of spectral weight transfer with increasing U/W in a typical Mott-Hubbard system (from the DMFT solution of the Hubbard model [A. Georges et al., Rev. Mod. Phys. 68, 13 (1996); G. Kotliar and D. Vollhardt, Phys. Today 57, 53 (2004)]) (ii). The figure and caption are adopted from H. F. Yang et al. [Phys. Rev. B 94, 115151 (2016)].

Fig. C9. Incoherent-to-coherent crossover in monolayer SRO. (a) Fermi surface maps of pristine and K-dosed monolayer SROs measured at 10 K. (b) Band distributions of pristine and K-dosed monolayer SROs along the $k_y = 0.2 \text{ \AA}^{-1}$ (red dotted line in (a)). (c, d) EDCs from pristine and K-dosed monolayer SRO films at the X and Γ points, normalized by $E = -0.6$ and -2 eV, respectively. With K dosing, a quasi-particle peak appears near the Fermi level and the hump peak in the high binding energy region disappears, as marked by the inverted triangle. 'K-dosed $\times 2$ ' indicates twice the dosing amount of 'K-dosed'.

REVIEWERS' COMMENTS

Reviewer #1 (Remarks to the Author):

Since the authors have addressed my concerns to the last version of this manuscript with satisfaction, I agree its publication in NC.

Reviewer #2 (Remarks to the Author):

The authors have done an excellent job of addressing my concerns and making appropriate modifications to the manuscript. I recommend publishing the revised manuscript in Nature Communications.

Reviewer #3 (Remarks to the Author):

The authors have answered all the questions I raised. I recommend the manuscript to publish in Nature Communications.